# IQ Was Not Improved by Post-Discharge Fortification of Breastmilk in Very Preterm Infants

**DOI:** 10.3390/nu14132709

**Published:** 2022-06-29

**Authors:** Anja Klamer, Line H. Toftlund, Kristjan Grimsson, Susanne Halken, Gitte Zachariassen

**Affiliations:** 1Hans Christian Andersen Children’s Hospital, Odense University Hospital, 5000 Odense, Denmark; kristjan.grimsson@rsyd.dk (K.G.); susanne.halken@rsyd.dk (S.H.); gitte.zachariassen@rsyd.dk (G.Z.); 2Department of Paediatrics, Holbaek Hospital, 4300 Holbaek, Denmark; l.toftlund@gmail.com; 3Faculty of Health Sciences, University of Southern Denmark, 5000 Odense, Denmark

**Keywords:** post-discharge nutrition, breastfeeding, fortification, preterm infant, cognitive development, Wechsler Intelligence Scale for Children IV, Five to Fifteen Questionnaire

## Abstract

(1) Very preterm infants are at increased risk of cognitive deficits, motor impairments, and behavioural problems. Studies have tied insufficient nutrition and growth to an increased risk of neurodevelopmental impairment; (2) Methods: Follow-up study on cognitive and neuropsychological development at 6 years corrected age (CA) in 214 very preterm infants, including 141 breastfed infants randomised to mother’s own milk (MOM) with (F-MOM) or without (U-MOM) fortification and 73 infants fed a preterm formula (PF-group), from shortly before discharge to 4 months CA. Infants with serious congenital anomalies or major neonatal morbidities were excluded prior to intervention. The Wechsler Intelligence Scale for Children IV was used for cognitive testing, and the children’s parents completed the Five to Fifteen Questionnaire (FTF); (3) Results: Post-discharge fortification of MOM did not improve either full-scale intelligence quotient (FSIQ) with a median of 104 vs. 105.5 (*p* = 0.29), subdomain scores, or any domain score on the FTF questionnaire. Compared to the PF group, the MOM group had significantly better verbal comprehension score with a median of 110 vs. 106 (*p* = 0.03) and significantly better motor skills scores on the FTF questionnaire (*p* = 0.01); (4) Conclusions: The study supports breastfeeding without fortification as post-discharge nutrition in very preterm infants, and it seems superior to preterm formula.

## 1. Introduction

Very preterm infants are at increased risk of poor neurodevelopmental outcome (cognitive deficits, motor impairments and behavioural problems) in childhood [1,2,3]. Protein intake and infant growth have been positively associated with larger volumes of various areas of the brain and higher cognitive and motor scores, respectively [4]. Most trials have focused on nutrition and growth during hospitalisation in the neonatal intensive care unit (NICU) where MOM and donor human milk (DHM) mixed with multi-nutrient fortifiers containing protein, carbohydrates, fat and various micronutrients are widely used to improve growth [5]. In formula-fed preterm infants, several studies have evaluated the importance of special preterm formulas containing higher levels of micro- and macronutrients compared to a term formula in order to improve growth and cognitive outcome [6,7]. In preterm infants, human breast milk has several advantages compared to formula. This is mainly due to a reduced incidence of necrotising enterocolitis (NEC), late onset sepsis (LOS), retinopathy of prematurity (ROP), and possibly also a reduction in bronchopulmonary dysplasia (BPD) [8,9]. Long-term studies of breastfeeding in both preterm and term-born infants show a small but significant positive effect on cognitive development at school age [10]. However, studies on post-discharge nutrition in preterm infants and later development are sparse. The majority of studies are observational and only very few RCT’s with early follow-up at 12–18 months exist. The observational studies rather consistently find a positive correlation between both pre- and post-discharge weight gain and head growth with later cognitive outcome in preterm infants [11]. Post-discharge, preterm breastfed infants demonstrate slower growth compared to formula-fed infants, indicating a potential negative effect of exclusive post-discharge breastfeeding on neurodevelopmental outcome [12]. Therefore, more knowledge on different types of post-discharge nutrition and later cognitive development is needed.

We hypothesised that continuous fortification added to MOM compared to a sole MOM-based diet from discharge to 4 months corrected age (CA) would improve cognitive and psychological development in very preterm infants at 6 years of age. Furthermore, we hypothesised that breastfeeding with or without fortification, compared to formula, would improve cognitive and psychological development.

## 2. Materials and Methods

We performed a cognitive and neuropsychological follow-up at 6 years of age in a cohort of consecutively included very preterm infants (gestational age (GA) ≤ 32 + 0 weeks) from four neonatal units in Denmark born in the period from July 2004 to August 2008, as described elsewhere [13]. From birth to post-menstrual age (PMA) of at least 30 weeks, infants were fed MOM or DHM. The fortification was initiated at 10–14 days of life and continued with decreasing amounts until discharge. DHM was replaced with a preterm formula from PMA 30 weeks if the available amounts of expressed MOM were insufficient [14]. Shortly before discharge, breastfed infants were randomised to a post-discharge diet of unfortified breastfeeding (U-MOM group) or addition of a fortifier (F-MOM group) until 4 months CA. Infants of non-breastfeeding mothers were forming the PF group. The intervention group F-MOM was daily given five packets of Enfamil^®^ Human Milk Fortifier (HMF) (Mead Johnson Nutritional, Evansville, IN, USA); 17.5 kcal and 1.375 g of protein per five packets. The PF group was given Enfalac Preterm Formula (Mead Johnson Nutritionals, Nijmegen, The Netherlands); 68 kcal, 2 g protein, 7.4 g carbohydrate, and 3.5 g fat per 100 mL. The details on nutrition and changes in diet during the intervention period have been described previously [13].

Infants without serious congenital or chromosomal anomalies, major neonatal morbidities, or severe eating disabilities at discharge were eligible for the study. Major neonatal morbidities were defined as periventricular leukomalacia or intraventricular hemorrhage grade 3 or 4 (according to Papille), BPD (any respiratory support at PMA 36 weeks), NEC, or cardiac malformation including persistent ductus arteriosus requiring surgery [13].

Infant growth with weight, length, and head circumference (HC) was measured at discharge, term, and at 2, 4, 6 and 12 months CA [13]. SGA was defined as BW less than −2 SDS (based on the reference Niklasson and Albertsson-Wikland 2008) [15]. From 2010 to 2015, the 6-year follow-up was conducted including an evaluation of growth, metabolism, lung development, and a psychological test battery of the Five to Fifteen (FTF) parent questionnaire and the Wechsler Intelligence Scale for Children IV (WISC-IV). The children were examined at the outpatient clinic at Odense University Hospital (OUH) on two different days (coordination of DXA, blood samples, body box, and WISC-IV). The outcome on growth, metabolism, and lung development is published elsewhere [16,17,18]. Of the 320 children enrolled in the original RCT, 214 children participated in the neuropsychological follow-up study at 6 years of age (Figure 1).

The WISC-IV is a test of intellectual ability (IQ). The test is divided into four domains: Verbal Comprehension (VC), measuring the child’s ability to comprehend and use acquired word knowledge and vocabulary; Perceptual Reasoning (PR), measuring the child’s ability to understand visuospatial information and use it to solve problems; Working Memory (WM), measuring the child’s ability to register, remember, and manipulate visual and auditory information; and Processing Speed (PS), measuring the child’s ability to quickly understand verbal or visual information and use it to solve a problem. The test results for each domain are transformed into an index score, and these indexes together provide the Full-Scale Intelligence Quotient (IQ). The WISC-IV was translated and validated in a representative Danish population of 477 children aged 6 to 16 years and 11 months in 2008–2010 and published for use in 2010. A psychologist at the Hans Christian Andersen Children’s Hospital blinded for the intervention, administered all of the WISC-IV tests and helped parents complete the FTF questionnaire. If a child was unable to complete the test on a specific day, another visit was scheduled if possible, unless the parents declined to participate further.

The FTF parent questionnaire evaluates neuropsychological and behavioural challenges with a focus on ADHD symptoms and comorbid conditions in children aged 5 to 15 years. The questionnaire was filled in by parents while their child completed the WISC-IV at the 6-year follow-up at OUH. The questionnaire intended for children under the age of 8 consists of 152 questions in the domains: memory, language, executive functions, motor skills, perception, social skills, and emotional/behavioural problems. The test describes both strengths and disabilities and is well accepted by parents [19]. Each question is answered with ‘Does not apply’ (0 point), ‘Applies sometimes or to some extent’ (1 point), or ‘Definitely applies’ (2 points). A mean score for each domain is calculated by adding the results and dividing by the total number of questions in that category [19,20]. The lower the score, the better the result. Scandinavian psychiatrists and psychologists developed the questionnaire in 2004 and standardised it in a normal population of Swedish children. Test results at or above the 90th percentile indicate a risk of developmental disorders and identify an area of difficulty for the individual child [21,22].

The mothers’ social group was defined as previously described in the original study; data were missing for two mothers in the U-MOM group [13]. Young mothers were defined as mothers under 25 years. The parents of five children filled in the FTF questionnaire less than 1 month before 6 years CA. These children were scored according to the 6–8-year normal range.

### Statistics

The SPSS package version 28 was used for statistical analyses. For comparison of means, the independent samples *t*-test or the Mann-Whitney U test was used. Fisher’s exact test was used to compare frequencies with small sample sizes, and one-way ANOVA was used for differences between means of more than two groups. Multiple linear regressions were used for continuous outcome variables, and binary logistic regression was used for the assessment of binary ordinal outcome variables. Odds ratios were calculated to interpret the results of the binary logistic regression analysis. We considered *p* values < 0.05 as significant. We predefined the following variables in the regression model: nutritional group (U-MOM, F-MOM, PF), gestational age (GA), birth weight standard deviation score (BW-z-score), sex (boys and girls), multiple birth (including triplets), and social status (low and high). Social status was originally divided into five groups ranging from one to five, with one being the highest social group. For the analysis, ‘social group’ was divided into high (one + two) and low (three + four + five). The results from the FTF questionnaire are presented as a dichotomous variable where the individual child is either under the 90th percentile (no concern) or at or above (concerned) according to table two in the user manual [23]. WISC-IV data are reported as mean (±SD) or median (range) according to normal distribution. All of the analyses are intention-to-treat analyses to ensure that the results can be applied in clinical practice.

## 3. Results

Out of 320 children in the original study, 66.9% participated in the neuropsychological follow-up (Figure 1). We found no significant difference in dropout rates or number of incomplete tests when comparing the three nutritional groups (data not shown). The psychologist obtained complete WISC-IV tests and FTF questionnaires from 206 and 205 children, respectively. The remaining children had incomplete test results (Figure 1).

We found no significant difference when comparing nutritional groups according to GA, BW, BW-z score, number of children born SGA, PMA at discharge, weight or weight z-score at discharge, or age at follow-up (Table 1).

We had significantly more boys in the PF group compared to the F-MOM group (*p* = 0.02) but not the U-MOM group (*p* = 0.1). Singletons were significantly more frequent in the U-MOM group compared to both the F-MOM and the PF groups (*p* < 0.01). In the PF group, more children were born to mothers from the low social group compared to both the U-MOM (*p* = 0.03) and the F-MOM group (*p* = 0.01). No significant differences in maternal age were observed between the groups. When the intervention ended at 4 months CA, the infants in the PF group were significantly heavier and longer compared to the infants in the U-MOM group (*p* = 0.02 and 0.03, respectively) but not the F-MOM group. No difference in HC SDS was observed at end of intervention. At 1 year CA, we found no significant differences in z-scores (weight, length and HC) between the groups.

### 3.1. Main Outcome on WISC-IV

#### 3.1.1. WISC-IV in the U-MOM Group vs. the F-MOM Group

In univariate analyses, we found no significant difference in IQ or the VC, PR, WM, or PS indexes when comparing the U-MOM and F-MOM groups (Table 2). IQ test scores in the subgroups were only approximately normally distributed due to outliers.

To control for the effect of other known factors that could possibly influence the IQ and subdomain scores, we conducted multiple regression analyses. The fortification of MOM did not significantly affect IQ scores. However, a high maternal social group increased the child’s IQ score with 7.0 points (3.6–10.3) compared to a low maternal social group. Singletons had higher IQ scores compared to multiple birth (5.6 points (1.8–9.4)), and one-week increase in GA increased the IQ with 1.1 (0.3–1.9) points. An increase of one in birth weight z-score increased the IQ with 1.6 points (0.1–3.2). Sex did not affect the IQ score. The VC index was significantly higher in children of mothers from a high social group (4.4 points; 1.2–7.6) or from singleton birth (4.8 points; 1.2–8.4). The PR index was also significantly higher in children of mothers from a high social group (8.5 points; 4.9–12.2), increasing GA (1.5 points pr. week; 0.6–2.4) or singleton birth (5.5 points: 1.4–9.6), whereas the WM index was significantly affected only by maternal social group (9.2 points; 5.2–13.1) and singleton birth (4.7 points; 0.2–9.2). The PS index was higher in girls (5.9 points; 1.3–10.6).

#### 3.1.2. WISC-IV in the MOM Group vs. the PF Group

As fortification did not affect the results of the WISC-IV test, we merged data from the two groups fed with MOM in the regression model for comparison with the PF group. Nutrition group did not significantly affect the IQ. However, higher GA, BW z-score, maternal social group, and singleton birth significantly increased the IQ score. An increase in GA with one week improved the IQ score by 1.5 (0.8–2.3) points, whereas an increase of one in BW z-score improved the IQ score by 2.0 (0.7–3.2) points. High maternal social group increased the IQ score by 6.2 (3.4–9.1) points. Singletons had an IQ of 6.2 (3.0–9.0) points higher compared to twins/triplets. Gender did not correlate with the IQ score.

Infants fed with MOM had a significantly higher VC index score with an increase of 3.2 (0.3–6.1) points compared to the PF group. GA (0.7 points per week; 0–1.4), BW z-score (1.5 points per increase of one z-score; 0.3–2.7), high social group (4.4 points; 1.6–7.1) and being born as a singleton (5.6 points; 2.8–8.5) also significantly augmented the VC index score in the regression model. In the PR index results, higher GA (1.5 points; 0.8–2.3), increase in BW z-score by one (1.7 points; 0.4–3.1), high social group (7.1 points; 4.1–10.1), and singleton birth (5.3 points; 2.1–8.4) significantly increased test scores. The WM index results were also significantly and positively correlated to GA (1.2 points per week; 0.3–2), BW z-score (2.0 points per increase of one; 0.5–3.4), singleton birth (3.5 points; 0.1–7), and high maternal social group (7.6 points; 4.3–10.9). The PS index results were significantly higher in girls compared with boys (6 points; 2.2–9.8) and increased with increasing GA (1.3 points per week; 0.4–2.3).

### 3.2. Five to Fifteen Questionnaire

#### 3.2.1. FTF in the U-MOM Group vs. the F-MOM Group

In univariate analyses, we found significantly more children with motor skills scores under the 90th percentile in the F-MOM group compared to the U-MOM group, 90.3% vs. 79.9%, respectively (*p* < 0.05) (Table 3). Fortification did not affect any of the other domain scores in univariate analyses.

In a binary logistic regression model including nutritional group, GA, BW z-score, social group, multiple birth, and gender, the effect of nutrition group on motor skills was no longer present. The nutrition group did not explain any of the variation in scores in the remaining subgroups when comparing the U-MOM and the F-MOM groups. However, children of mothers in the low social group had a significantly increased risk of a test score at or above the 90th percentile in the domains: executive functions (OR: 4.3; 1.1–16.7), perception (OR: 2.9; 1.2–6.9), memory (OR:2.5; 1.1–6.1), language (OR: 3.2; 1.0–9.9), social skills (OR: 4.5; 1.7–11.8), and emotional/behavioural problems (OR: 7.4; 2.4–23.3).

Boys had a higher risk of performing at or above the 90th percentile in the domains: motor skills (OR: 4.5; 1.5–13.7), executive functions (OR: 5.6; 1.4–21.3), perception (OR: 3.6; 1.5–8.5), memory (OR: 3.0; 1.3–7.2), language (OR: 3.0; 1.0–8.9), and social skills (OR: 2.8; 1.2–6.9), but not emotional/behavioural problems compared to girls. For the memory domain, one decrease in BW z-score increased the odds ratio of a score at or above the 90th percentile with 1.8 (1.1–2.7).

#### 3.2.2. FTF in the MOM Group vs. the PF Group

For comparison with the PF group, we merged the two MOM groups as the results from the FTF questionnaire in these groups did not differ. In the univariate analyses, the PF group had significantly more children at or above the 90th percentile in motor skills (*p* < 0.01), perception (*p* = 0.02), language (*p* = 0.01), and social skills (*p* = 0.01). In the regression analyses, the PF group still had a significant increase in odds ratio for scores at or above the 90th percentile in motor skills 2.6 (1.2–5.5) (*p* = 0.012). Boys had a significantly increased risk of a domain score at or above the 90th percentile in motor skills (OR = 2.7 (1.3–5.8)), executive function (OR = 5.3 (1.7–16.7)), perception (OR = 4.0 (2.0–7.8)), memory (OR = 2.3 (1.2–4.4)), language (OR = 3.2 (1.4–7.5)), and social skills (OR = 2.2 (1.1–4.3)), but not emotional/behavioural problems.

Belonging to the low social group significantly increased the risk of a score at or above the 90th percentile in social skills (OR = 2.1 (1.0–4.3)) and behavioural problems (OR = 3.3 (1.5–7.2)). Singletons had a significantly reduced risk of a score at or above the 90th percentile in perception (OR = 0.4 (0.2–0.9)), but a one week decrease in GA significantly increased the risk of a score at or above the 90th percentile in perception (OR = 1.2 (1–1.43)).

## 4. Discussion

To our knowledge, this is the first study evaluating the effect of post-discharge fortification added to MOM on cognitive outcome at school age.

Post-discharge fortification of MOM in our population of very preterm infants without severe neonatal morbidity did not affect the IQ score at 6 years of age. This result is in accordance with the studies of Aimone et al., and da Cunha et al. [24,25], who found no difference in Bayley test scores at 1 year. However, Aimone et al. found a trend towards better language and motor skills in the supplemented group, which also had significantly better growth at 1 year of age. Da Cunha et al., found a trend towards better domain scores, especially in the language domain, in infants fed with F-MOM compared to U-MOM. We did not find the same trend in school-aged children between our MOM groups, however, the MOM group had higher VC index compared to the PF group.

We only provided an extra of 1.375 g of protein added to a small amount of MOM during breastfeeding due to ethical concerns related to the risk of breastfeeding discontinuation. We observed no difference in anthropometry at 4 months CA between the MOM groups. It is possible that a larger amount of protein would have changed the result. Dabydeen and colleagues found that preterm infants with perinatal brain damage receiving 120% of the recommended average intake of protein and energy during the first year of life had larger HC and corticospinal tract diameter at 12 months of age [26]. Even though our intervention ceased at 4 months CA and only included infants without major neonatal morbidities, we did not observe the same effect on HC growth in our F-MOM group. Further studies are needed to elucidate whether fortification of breast milk with extra protein and calories improves cognitive outcome in infants with more morbidities or impaired growth during hospitalisation. Among infants with NEC excluded from our RCT, Honore et al. found HC to be significantly smaller up to 6 month CA [27], stressing the importance of future studies to include groups of severely ill very preterm infants.

Our original study population was consecutively recruited and thereby represents the ‘general’ population of healthy very preterm infants. We included 69% of the original study population, and dropout rates did not differ significantly between the follow-up groups. Hence, we consider our results to be representative of the entire study group.

The inclusion of a PF group is a strength in our study, yielding the opportunity to compare cognitive outcome in formula- and breast-fed infants, even though our PF group was not randomised, since infants cannot be randomised to breastfeeding or formula feeding for ethical reasons. As expected, the PF group included more mothers with a low socioeconomic status and twins/triplets, but also by chance more boys, which we controlled for in the regression analysis. We did not find an effect of breastfeeding on IQ as Lucas et al. found 30 years ago [28]. However, the VC index results were significantly higher in the MOM group compared to the PF group by 3.2 points corresponding to a two-week increase in GA, even when controlled for confounders. We find this result very interesting, and language (as a proxy for brain development) may be a particularly sensitive area to nutritional intervention [7,24,25]. Nevertheless, our results need to be confirmed by larger intervention studies.

As expected, IQ scores were positively correlated with high social group, singleton status, increasing GA, and BW z-score. Our results are thus in accordance with findings from other studies [1,29]. In the WISC-IV test, girls performed better in the Processing Speed index but not in IQ scores. This result is in accordance with other studies finding boys to be more challenged by preterm birth [30]. Our IQ scores were very high in the entire group considering the preterm birth. Due to study design limitations, we did not include a term-born control group. However, in the ‘Odense Child Cohort’ [31], children had a mean IQ score at 7 years of 99.1 (SD 11.9) on the WISC-V [32], and even though our study cohort was defined by the absence of major neonatal morbidities, we expected IQ scores to be below average due to prematurity [1]. As our entire cohort was examined by one psychologist, we expect the comparison between the groups to be unaffected by the rather high mean score on the WISC-IV.

Parts of the FTF questionnaire have previously proven useful in evaluating a group of extremely preterm born children [33], but to our knowledge, this is the first time the entire questionnaire is used in a preterm population. Our results thus contribute to the very sparse literature on post-discharge nutrition and psychiatric challenges. As preterm infants are known to be at increased risk of psychiatric disabilities [34,35], we find this FTF questionnaire to be a valuable tool to describe a broad range of the psychological challenges among this group.

In a large meta-analysis, motor impairment was reduced in low birth weight (LBW) infants receiving supplementation of MOM in the hospital compared to children receiving no supplementation but with no effect of supplementation post-discharge or both pre- and post-discharge [36]. Our results in the MOM group are in accordance with these findings as our entire group received fortification during hospitalisation. However, a diet based on breastfeeding had a significantly positive effect on motor skills compared to infants fed with PF post discharge. As motor impairment is a frequent finding in the very preterm population [37], our results highlight the importance of breastfeeding. Further studies are needed to evaluate the effect of post-discharge breastfeeding in infants with more neonatal morbidities than our cohort.

It was surprising that only 8.5–14.5% of children had test results above the 90th percentile in executive functions, as this domain score includes symptoms of ADHD, and the FTF questionnaire has previously been found very useful at detecting ADHD symptoms in children [19]. This finding may be due to the lack of major neonatal morbidities in our study group and hence the lower risk of brain damage [2].

Boys had significantly poorer scores than girls in all domains except in emotional/behavioural problems. This is in accordance with Bohlin et al., who found boys to have higher scores in the domains: executive function, motor skills, perception, and social skills [20]. Lambek et al., found sex to have less influence on the test results. However, their results were from a cohort examined for psychiatric evaluation, where girls were expected to have more symptoms [38]. The new edition of the FTF questionnaire has different norm values for boys and girls.

GA, BW z-score, and singleton birth did not substantially affect the FTF scores in our cohort. This finding could be due to selection bias as our cohort excluded infants with major neonatal morbidities, which are seen more frequently in the more immature infants, twins, and infants with SGA [39]. Perhaps the spectrum of BW z-scores and GA was too narrow, resulting in a cohort too homogenous to yield significant results. Alternatively, the results could be because the FTF is not a diagnostic tool but only used for screening. Compared to a normal sample with expected 10% to be at or above the 90th percentile, the results do, however, indicate more problematic behaviour and challenges in everyday tasks in our preterm cohort. This has also been supported by other studies [33,40]. In all areas except executive functions, our study cohort had more than 10% of children with a score indicating difficulties in a particular area. This finding supports the theory of a ‘preterm behavioural phenotype’ [2] and stresses the importance of long-term follow-up in this population as well as the possible need for supportive care in everyday activities among peers.

A limitation is that our study was not powered to detect differences in IQ or FTF at 6 years of age due to study design and inclusion for a limited time period.

However, our data may assist clinicians working in the difficult field of nutritional recommendations on the discharge of very preterm infants in advocating breastfeeding and supporting long-term follow-up.

## 5. Conclusions

In conclusion, our study contributes to the knowledge on the benefits of a MOM-based diet post-discharge since previously breastfed children had better verbal comprehension index on WISC-IV and better motor skills score on the FTF questionnaire compared to PF fed children, even when the results were adjusted for known confounders.

## Figures and Tables

**Figure 1 nutrients-14-02709-f001:**
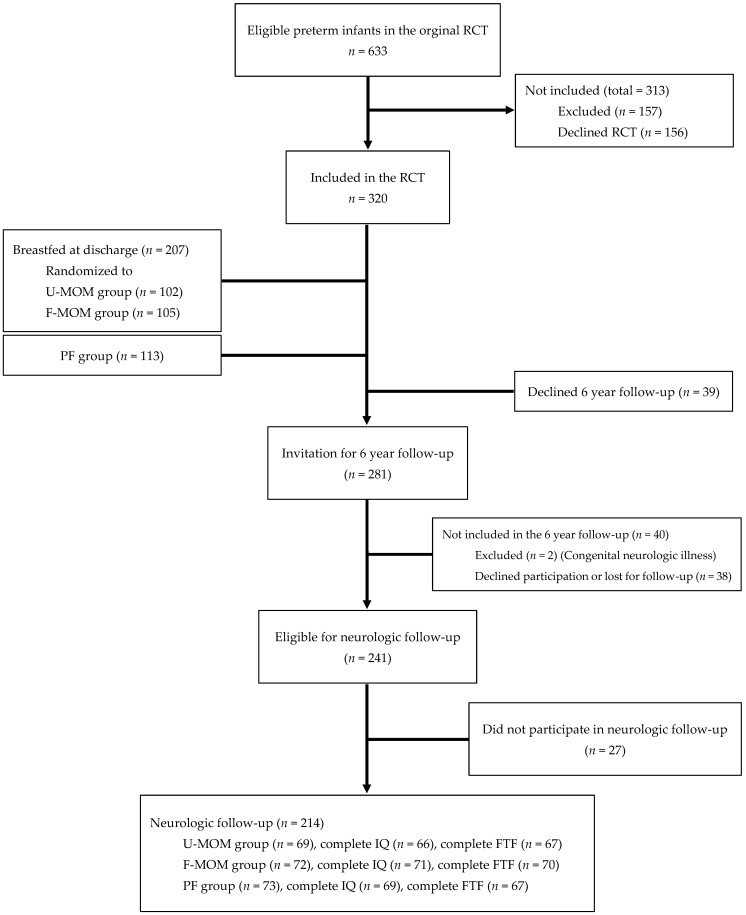
Flowchart of the study population.

**Table 1 nutrients-14-02709-t001:** Basic Characteristics of the study population.

	Unfortified Mother’s Milk (U-MOM)	Fortified Mother’s Milk (F-MOM)	Preterm Formula (PF)
Preterm infants	69	72	73
Boys % (*n*)	49.3% (34)	43.1% (31)	63% (46)
Singletons % (*n*)	82.6% (57)	61.1% (44)	54.7% (40)
GA at birth (Weeks + days) (Median; min–max)	29 + 4 (24 + 1–32 + 0)	30 + 2 (24 + 3–32 + 0)	29 + 4 (26 + 2–31 + 5)
Birth weight (grams) (Mean ± SD)	1284 ± 376	1318 ± 335	1280 ± 339
Weight z-score (SDS) at birth (Mean ± SD)	−0.96 ± 1.12	−1.04 ± 1.04	−1.25 ± 1.14
SGA % (*n*)	18.8% (13)	19.4% (14)	26% (19)
PMA at discharge (Weeks and days) (Median; min–max)	37 + 3 (34 + 5–45 +4)	37 + 2 (34 + 5–42 + 3)	36 + 6 (35 + 0–40 + 6)
Weight at discharge (g) (Mean ± SD)	2679 ± 469	2612 ± 343	2696 ± 427
Weight z-score (SDS) at discharge (Mean ± SD)	−1.19 ± 0.87	−1.20 ± 0.83	−0.96 ± 0.95
Head circumference z-score (SDS) at 4-month CA (Mean ± SD)	0.24 ± 0.94	0.26 ± 1.08	0.44 ± 1.29
Weight z-score (SDS) at 4-month CA (Mean ± SD)	−0.65 ± 1.19	−0.54 ± 1.20	−0.17 ± 1.17
Length z-score (SDS) at 4-month CA (Mean ± SD)	−0.36 ± 1.42	−0.14 ± 1.34	0.16 ± 1.41
Corrected age (years) at WISC-IV (Median; min-max)	6.4 (5.4–8.3)	6.4 (5.2–7.5)	6.3 (5.9–7.7)
Corrected age (years) at FTF questionnaire (Median; min–max)	6.5 (6.1–8.3)	6.4 (6.0–7.5)	6.3 (5.9–7.7)
Maternal age (years) (Mean ± SD)	30.5 ± 4.6	31.3 ± 4.5	30.9 ± 5.4
Young mother % (*n*) *	8.7% (6)	5.6% (4)	12.3% (9)
Maternal low social group (group 3–5) % (*n*)	53.7% (36)	51.4% (37)	71.2% (52)

* defined as younger than 25 years.

**Table 2 nutrients-14-02709-t002:** Results of the WISC-IV test.

	U-MOM Group	F-MOM Group	MOM Group	PF Group
Full Scale IQ (Median; min–max)	105.5 (63–122)	104 (64–132)	105 (63–132)	104 (71–127) *
Verbal Comprehension index (Median; min–max)	112 (83–130)	110 (63–130)	110 (63–130)	106 (73–128) *
Perceptual Reasoning index (Median; min–max)	109 (62–128)	105 (77–141)	107 (62–141)	101 (69–126) *
Working Memory index (Median; min–max)	98 (60–122)	98 (63–116)	98 (60–122)	95 (57–116)
Processing Speed index (Median; min–max)	104 (69–133)	104 (75–141)	104 (69–141)	104 (69–136)

* sig. lower than MOM group.

**Table 3 nutrients-14-02709-t003:** Results of the Five to Fifteen Parent Questionnaire.

Skill	Percentile	Post Discharge Nutrition Group			
		U-MOM Group	F-MOM Group	MOM Group	PF Group
Motor skills (*n*)	≥90th <90th	22.1% (15) 77.9% (53)	9.7% (7) 90.3% ^ (65)	15.7% (22) 84.3% * (118)	33.8% (24) 66.2% (47)
Executive functions (*n*)	≥90th <90th	14.5% (10) 85.5% (59)	8.5% (6) 91.5% (65)	11.4% (16) 88.6% (124)	12.7% (9) 87.3% (62)
Perception (*n*)	≥90th <90th	42.4% (22) 67.6% (46)	22.2% (16) 77.8% (56)	27.1% (38) 72.9% * (102)	43.7% (31) 56.3% (40)
Memory (*n*)	≥90th <90th	28.4% (19) 71.6% (48)	22.2% (16) 77.8% (56)	25.2% (35) 74.8% (104)	35.7% (25) 64.3% (45)
Language (*n*)	≥90th <90th	16.4% (11) 83.6% (56)	12.5% (9) 87.5% (63)	14.4% (20) 85.6% * (119)	29.4% (20) 70.6% (48)
Social skills (*n*)	≥90th <90th	29% (20) 71% (49)	16.7% (12) 83.3% (60)	22.7% (32) 77.3% * (109)	39.1% (27) 60.9% (42)
Emotional/behavioural problems (*n*)	≥90th <90th	23.9% (16) 76.1% (51)	18.1% (13) 81.9% (59)	20.9% (29) 79.1% (110)	31.9% (22) 68.1% (47)

* *p* < 0.05 MOM diet vs. preterm formula, ^ *p* < 0.05 unfortified MOM vs. fortified MOM.

## Data Availability

Data can be obtained from the corresponding author.

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
