# Peer review of "IQ Was Not Improved by Post-Discharge Fortification of Breastmilk in Very Preterm Infants"

_nutrients, 2022, doi:10.3390/nu14132709_

Round 1

Reviewer 1 Report

Nice to have a RCT that follows neurodevelopment out so far, decent follow-up percentage. Has the potential to fill a void in the literature. Overall the article was well written and nicely done. 

Biggest complaint was that the title of the article is not well supported by the data and conclusions drawn within the article. The focus of the article was really the difference between MOM and PF groups. The lack of differentiation between fortified and unfortified MOM was not a significant part of the article. Further, there were significant differences between the MOM and PF groups that are known contributors to IQ, making it difficult to truly assess the impact of the diet on outcomes. Indeed, the significantly increased percentage of singletons and higher social group may have overcompensated for the unfortified breast milk.

Results are also difficult to extrapolate, given the infants who are highest risk for neurodevelopmental impairment (those with complications during NICU stay including BPD and IVH) were excluded from this study. These infants who have been sicker during the hospitalization are the ones more likely to suffer a higher degree of extrauterine growth restriction, and would potentially benefit the most from fortification. Would include some of these things within the limitations in the study.

On the study population flowchart, the number of u-MOM that completed neurologic follow-up is noted as 69. But the box below states that 70 completed the FTF. One of these numbers is incorrect.

Author Response

Thank you very much for your comments. 

  1. We find the title to support the original RCT on effect of fortification of MOM. A tilte, including both findings of fortification not being better than MOM, and MOM being better than PF, would be to long, in our opinion.
  2. We find the focus of the article to be on the lack of difference between the MOM groups, which was the primary focus for the RCT. However as we found no difference between groups and our study is the first to follow children to the age of 6, we can only draw the conclusion that fortification does not improve IQ. Due to the lack of difference between the MOM groups, they were merged for comparison with the the PF group. Hence the conclusion supporting MOM without fortification and an overall support of breastfeeding compared to PF.  
  3. Our randomization did not take social status and singleton status into account, and therefore we only report few results from univariate analysis. However both social status and singleton status are included in the regression analyses due to their known influence on IQ-outcome. 
  4. Our study does not include infants with major neonatal morbidities as described in the article. Future studies should focus on these groups of infants using the infants without major morbidities as a reference. We have stressed this in the disucssion and added a reference to another study on growth in infants that were excluded from this RCT due to NEC. 
  5. The result of 70 is a typing error, the correct number is 67. Thank you very much for drawing our attention to this. 

Reviewer 2 Report

This study contributes with an interesting knoledge on benefits of breastfeeding  on neurodevelopmental capacities of very preterm infants. 

The paper has a good quality in execution and presentation. Only a few remarks could be made:

*Figure 1: last square "Neurologic follow-up": Preterm formula?

*Table 3: MM?, better "MOM"

Line 304: EP ? 

References: decide if the title of the articles goes on capitals or not.

Author Response

Thank you very much for your comments.

  1. PF group = Preterm Formula group. We have enlarged the Neurologic Follow-up heading in the first line of the square.
  2. MM in table 3 has been corrected to MOM. 
  3. EP has been changed to extremely preterm born. 
  4. For references we have used Endnote X9 TF-Standard NLM and all references should be cited accordingly.